# Relationship between Maternal Fever and Neonatal Sepsis: A Retrospective Study at a Medical Center

**DOI:** 10.3390/biomedicines10092222

**Published:** 2022-09-07

**Authors:** Sheng-Hua Huang, Yu-Jun Chang, Lih-Ju Chen, Cheng-Han Lee, Hsiao-Neng Chen, Jia-Yuh Chen, Chien-Chou Hsiao

**Affiliations:** 1Department of Neonatology, Changhua Christian Children’s Hospital, Changhua 50050, Taiwan; 2Big Data Center, Changhua Christian Hospital, Changhua 50050, Taiwan; 3Department of Post-Baccalaureate Medicine, College of Medicine, National Chung Hsing University, Taichung 40705, Taiwan; 4School of Medicine, Kaohsiung Medical University, Kaohsiung 80708, Taiwan

**Keywords:** maternal fever, neonatal sepsis, risk factor

## Abstract

Various risk factors are associated with neonatal sepsis; however, its relationship to maternal postpartum fever is unknown. This study aimed to determine the relationship between maternal postpartum fever and neonatal sepsis. Full-term and late preterm stable infants born from January 2019 to June 2021 and whose mothers developed intra- or post-partum fever were included in the study. After the newborns were transferred to the nursery, laboratory assessments were performed. Based on clinical conditions and data, the newborns were divided into unlikely sepsis and probable/proven sepsis groups. Maternal fever onset, duration, and maximum body temperature were recorded. We included 1059 newborns whose mothers developed fever intra-partum (*n* = 192), post-partum (*n* = 844), and intra- and post-partum (*n* = 23). The newborns were grouped into those with unlikely sepsis (*n* = 550) and those with probable/proven sepsis (*n* = 509). The incidence of intrapartum fever was higher in the probable/proven sepsis group than in the unlikely sepsis group (27.9% vs. 13.3%, *p* < 0.001). The incidence of postpartum fever was lower in the probable/proven sepsis group than in the unlikely sepsis group (74.7% vs. 88.5%, *p* < 0.001). Development of maternal fever within 1.8 h postpartum and a newborn respiratory rate of >60 breaths/min were positive predictors (91.6%) for neonatal probable/proven sepsis.

## 1. Introduction

Neonatal sepsis is an important cause of morbidity and mortality. It ranks third among the causes of neonatal death, following prematurity and intrapartum-related complications [1,2]. Based on the timing of presentation, neonatal sepsis is classified as early or late onset. Early-onset neonatal sepsis presents clinical manifestations within the first 3 days of life (<72 h). Some researchers consider early-onset neonatal sepsis to occur in the first 7 days of life [3]. The incidence of early-onset neonatal sepsis ranges from 1 to 5 per 1000 live births. It has been shown to decrease with intrapartum antibiotic therapy [4,5].

Early-onset infection is usually due to vertical transmission by ascending contaminated amniotic fluid or during vaginal delivery from bacteria in the mother’s lower genital tract [6]. Various maternal risk factors are related to early-onset neonatal sepsis, and intrapartum maternal fever is one of them [6,7,8,9,10]. However, no previous study has explored the relationship between maternal postpartum fever and neonatal sepsis. The time of maternal fever onset, including intrapartum and postpartum, may indicate a variation in the risk of neonatal sepsis. Thus, evaluating neonatal sepsis in cases with maternal postpartum fever may be warranted. Therefore, this study aimed to determine the relationship between maternal postpartum and intrapartum fever and early-onset neonatal sepsis. Other relevant factors are compared with neonatal sepsis, including the time point of the maternal fever (intrapartum and postpartum), fever duration, maximum body temperature, delivery method, presence of chorioamnionitis, premature rupture of membrane (>18 h), misoprostol use, epidural analgesia use, and Apgar score.

## 2. Materials and Methods

### 2.1. Study Population

Owing to the retrospective nature of the study, the need for informed consent was waived.

We collected data on infants born at Changhua Christian Children’s Hospital Changhua, Taiwan from January 2019 to June 2021. The babies were born at more than 35 weeks gestation and weighed >2200 g. They were in a relatively stable condition at birth. Their mothers developed fever (body temperature ≥ 38 °C) during or after delivery. The mothers’ information was also collected at birth.

### 2.2. Data Collected from the Newborns

The newborns were transferred to the nursery for blood sampling after birth. Complete blood count and differential count, C-reactive protein measurement, and blood culture were performed. The sample volume of the blood culture was 1 mL [6]. Other collected data were sex, gestational age, birth weight, Apgar score at birth, time from birth to blood sampling, and vital signs including body temperature, heart rate, and respiratory rate.

### 2.3. Data Collected from the Mothers

The data collected from the mothers included age, parity, delivery method, prenatal group B streptococcus test result, premature rupture of membrane (PROM) (<18 h or ≥18 h), misoprostol use (to induce labor before childbirth or to stop bleeding after childbirth), epidural analgesia use, presence of chorioamnionitis, the time point of the maternal fever (before or after child birth), duration of maternal fever, and highest maternal body temperature. The definition of intrapartum in our study is from the onset of labor through the delivery of the baby. The definition of postpartum in our study is from birth to 48 h after birth. The fever duration is defined as the time when the body temperature rose above 38 °C to the time when the body temperature dipped below 38 °C. If fever subsided for 24 h, and then flared up again, it will be defined as two fever episodes. We choose the longer time as our fever duration. If one is intrapartum fever and the other is postpartum fever, both will be listed in our data. The definition of chorioamnionitis is maternal fever > 38 °C (100.4 degrees F) plus ≥ 1 of the following conditions, according to the American College of Obstetricians and Gynecologists (ACOG) recommendations:

Maternal tachycardia (pulse > 100 beats/minute for 5 min);

Fetal tachycardia (heart rate ≥ 160 beats/minute for 5 min);

Uterine tenderness;

Foul-smelling amniotic fluid;

Maternal leukocytosis (> 15,000–18,000 cells/mm^3^).

Intrapartum antibiotics administrated due to a positive or uncertain GBS screen result; Screen result and intrapartum fever was recorded.

### 2.4. Diagnosis

Based on their clinical symptoms and blood test results, we divided the newborns into two groups: those with unlikely sepsis and those with probable or proven sepsis. A positive blood culture result indicated proven sepsis, whereas a negative finding with clinical symptoms (e.g., ongoing temperature instability, shortness of breath, chest wall retraction, desaturation, or neurologic symptoms not explained by other conditions) or abnormal laboratory data findings (white blood cell count <5000/μL or white blood cell count 6 h after birth >32,000/μL, absolute neutrophil count <1000/μL, C-reactive protein >1 mg/dL, ratio of immature to total neutrophil count >20% [11,12,13,14,15,16]) indicated probable sepsis. The rest of the newborns were included in the unlikely sepsis group.

### 2.5. Statistical Analysis

Continuous variables are presented as the median and interquartile range (25th–75th percentile), and categorical variables are presented as number and percentage. The Mann–Whitney U, chi-square, or Fisher’s exact tests were performed, as appropriate, to determine whether variables were different between the two groups (unlikely and probable/proven sepsis). A receiver operating characteristic (ROC) curve analysis was conducted to evaluate the diagnostic performance of the first onset of maternal fever and the infant’s respiratory rate in predicting neonatal probable/proven sepsis, and the Youden index was used to determine the cutoff point. Bivariable and multivariable logistic regression analyses were performed to identify potential risk factors for neonatal probable/proven sepsis. All data were analyzed using SPSS for Windows (version 22.0, IBM Corp., Armonk, NY, USA). Statistical significance was set at *p* < 0.05.

## 3. Results

A total of 1059 newborns born in a hospital between January 2019 to June 2021 were included. The mothers developed fevers before delivery (*n* = 192), after delivery (*n* = 844), and before and after delivery (*n* = 23). Table 1 shows the two sample groups; namely, unlikely sepsis (*n* = 550) and probable/proven sepsis (*n* = 509). The Mann–Whitney U test was used to compare the differences between the two groups. The factors that showed significant differences in distribution between the two groups were parity, time from postpartum misoprostol administration to fever onset, time from birth to fever onset, time point of the first maternal fever, gestational age, birth weight, 1 min Apgar score, time of data from infant birth, infant heart rate, and infant respiratory rate. Among the 1059 newborns, four had bacteremia. The time points of the maternal fever and microorganism confirmation are shown in Table 2. Among the newborns with bacteremia, three had mothers who developed fever after delivery, and two had blood cultures positive for *Escherichia coli* and *Streptococcus agalactiae*, which are commonly associated with early-onset neonatal sepsis [17,18,19].

Table 3 shows the relationship between perinatal factors and neonatal sepsis. We used the chi-square or Fisher’s exact tests to compare the differences between the two groups. The incidence of risk factors in the probable/proven sepsis group was higher than that in the unlikely sepsis group; the results will provide relevant figures from the probable/proven sepsis and unlikely sepsis groups, respectively. The number of parities was mostly one (76.2% vs. 66.4%, *p* = 0.004), and the delivery method was mainly vacuum extraction delivery (VED) (60.7% vs. 46.7%, *p* < 0.001). The time point of misoprostol use was before delivery (34.8% vs. 25.6%, *p* = 0.005), and the rate of epidural analgesia use was higher (82.1% vs. 60.4%, *p* < 0.001). Moreover, less prematurity (6.3% vs. 13.5%, *p* < 0.001), more newborns with premature rupture of membranes over 18 h (8.4% vs. 3.1%, *p* < 0.001), and more newborns with 1 min Apgar score <7 (2.6% vs. 0.5%, *p* = 0.01) were observed. The incidence of intrapartum fever (38–38.9 °C) (27.9% vs. 13.3%, *p* < 0.001) and high fever (≥39 °C) (5.7% vs. 2.2%, *p* = 0.003) was higher in the probable/proven sepsis group than that in the unlikely sepsis group. Postpartum fever was more common in the unlikely sepsis group (88.5% vs. 74.7%, *p* < 0.001).

Table 4 shows the findings of the logistic regression analysis that was used to evaluate the correlation between perinatal factors and neonatal sepsis. In the bivariable analysis, probable/proven sepsis was significantly associated with the following perinatal factors: less parity (crude odds ratio = 0.681, 95% confidence interval (CI) = 0.553–0.839, *p* < 0.001), misoprostol use before delivery (crude odds ratio = 1.562, 95% CI = 1.166–2.094, *p* = 0.003), epidural analgesia use during delivery (crude odds ratio = 3.111, 95% CI = 2.333–4.149, *p* < 0.001), VED as main delivery method (crude odds ratio = 1.632, 95% CI = 1.177–2.262, *p* = 0.003), 1 min Apgar score <7 (crude odds ratio = 4.779, 95% CI = 1.354–16.869, *p* = 0.015), and PROM >18 (crude odds ratio = 2.893, 95% CI = 1.628–5.142, *p* < 0.001). The risk of neonatal sepsis was significantly lower for women with postpartum fever than for women with intrapartum fever (crude odds ratio = 0.376, 95% CI = 0.270–0.523, *p* < 0.001). After adjusting for possible confounding factors, the multivariable analysis still showed that maternal postpartum fever had a significantly lower risk of neonatal sepsis (adjusted odds ratio = 0.661, 95% CI = 0.439–0.994, *p* = 0.047).

We used an ROC curve to evaluate the diagnostic performance of the first maternal fever time point and the infant’s respiratory rate in predicting neonatal probable/proven sepsis. ROC analysis showed that the areas under the curve were 0.639 (95% CI = 0.605–0.672, *p* < 0.001) for first maternal fever time point alone, and 0.829 (95% CI = 0.804–0.853, *p* < 0.001) for first maternal fever time point combined with infant’s respiratory rate (Figure 1). When the time point of the first maternal fever was within 1.8 h postpartum and the newborn’s respiratory rate was >60 breaths/min, the positive predictive value was 91.6% for neonatal probable/proven sepsis.

## 4. Discussion

Our results showed that newborns whose mothers developed postpartum fever were at a significantly lower risk of acquiring neonatal sepsis compared with those whose mothers developed intrapartum fever. It is clear that not all fevers of women in labor or postpartum are caused by microbial organisms. We also found that when the mother developed fever within 1.8 h postpartum and the baby exhibited rapid breathing after birth (>60 breaths/min), the positive predictive value for neonatal sepsis was 91.6%. By contrast, in the newborns in the unlikely sepsis group, 97.5% (specificity) of the mothers developed the first fever 1.8 h postpartum or the neonatal respiratory rate was ≤60 breaths/min. The correlation between maternal postpartum fever and neonatal sepsis has not been reported. In our proven sepsis case, the mother of the newborn developed fever after delivery. The case exhibited *E. coli* and *S. agalactiae* infections, which are the most common causes of early-onset sepsis [17,18,19]. The mother of the newborn with *E. coli* infection was diagnosed with chorioamnionitis, and the endometrial culture was positive for *E. coli*. However, the mother did not develop fever until after childbirth. If a mother has an infection, it can be passed on to the newborn before the onset of the fever. Therefore, if a mother has a fever after childbirth, the newborn may still have been exposed to the infection. According to the CDC guidelines, “in an effort to avert neonatal infections, maternal fever alone in labor may be used as a sign of chorioamnionitis and the newborn should undergo a limited evaluation and receive antibiotic therapy pending culture results” [20]. Numerous studies have found a correlation between maternal intrapartum fever and neonatal sepsis [6,7,8,9,10]. Our statistical results showed no correlation between intrapartum fever duration and neonatal sepsis [9]; however, a significant correlation between high intrapartum fever (≥39 °C) and neonatal sepsis was observed, which is consistent with the results of other studies [9,10].

Many studies have reported that preterm infants have a higher risk of neonatal sepsis than full-term infants [21,22], and the risk of infection in newborns with low birth weight is also increased [21,22]. However, in our study, preterm infants were not associated with neonatal sepsis, and those with lower birth weight were instead included in the unlikely sepsis group. The possible reason for this may be because the included newborns were all >35 weeks old, weighed >2200 g, and were relatively stable at birth. The possibility of infection is judged based on the follow-up clinical manifestations and laboratory findings. Newborns with low gestational age and low birth weight may have been admitted to general wards or intensive care units at the beginning and thus were not included in our study, which probably caused the difference in our study results.

Our study showed that epidural analgesia is associated with neonatal sepsis, but not causal effect. Some studies reported that epidural analgesia and intrapartum fever are significantly correlated [23,24,25]. Although the patients included in our study were all mothers with intrapartum or postpartum fever, our results showed that intrapartum fever was associated with neonatal sepsis, which may indirectly indicate a correlation between epidural analgesia and neonatal sepsis [25]. The obstetrics and gynecology department of our hospital uses misoprostol to induce labor or treat postpartum hemorrhage. A common adverse effect of misoprostol is the onset of maternal fever [26,27]. Our research recorded the time point of misoprostol use, and statistical results showed that the timing of misoprostol for labor induction is related to neonatal infection, which may be associated with the relatively unstable condition of the mother or fetus.

Our study showed that nulliparous mothers were associated with neonatal sepsis more than multiparous mothers, and the literature showed that nulliparous women are associated with intrapartum fever. A possible explanation for fever among nulliparous mothers could be the higher energy expenditure associated with muscle contraction [28]. Therefore, we included more nulliparous women in our evaluation. Other studies have reported a higher rate of nulliparous women who received epidural analgesia [29]. Similar to intrapartum fever and epidural analgesia, nulliparity was also associated with neonatal sepsis in our study. Previous studies emphasized that in mothers with fever, instrumental and cesarean deliveries comprised the higher proportion of delivery methods [9,28]. Our study pointed to a significant association between VED and neonatal sepsis, possibly explained by a less smooth birth process or a less stable fetus, which may be due to infection. Premature rupture of the membranes (>18 h) was significantly associated with neonatal sepsis in our study, which is consistent with the results from other reports [6,7,8]. Maternal chorioamnionitis is a well-recognized risk factor for early-onset neonatal sepsis [7,30]. Chorioamnionitis and neonatal infection did not reach a statistical correlation in our study. This may be because neonates whose mothers were diagnosed with chorioamnionitis were directly admitted to the ward and were not included in our study because of the unstable clinical status of the infection at birth. Babies with a low 5 min Apgar score were at risk of acquiring neonatal sepsis [31,32]. The newborns we included were relatively stable at birth, with a 5 min Apgar score >7 after management. A low 1 min Apgar score was associated with neonatal sepsis in our study. This may be because newborns at risk of infection are born in relatively unstable condition, without management.

The limitation of this study is that we only included neonates in stable condition at birth. Patients who were directly admitted to the hospital in unstable condition should have been included for comparison.

## 5. Conclusions

Infants whose mothers’ first fever occurs within 1.8 h postpartum and whose respiratory rate is >60 times/min may be at risk for sepsis and may require further examination. By contrast, infants whose mothers’ first fever begins >1.8 h postpartum or whose respiratory rate is ≤60 times/min may be less likely to develop sepsis and may not need aggressive intervention.

## Figures and Tables

**Figure 1 biomedicines-10-02222-f001:**
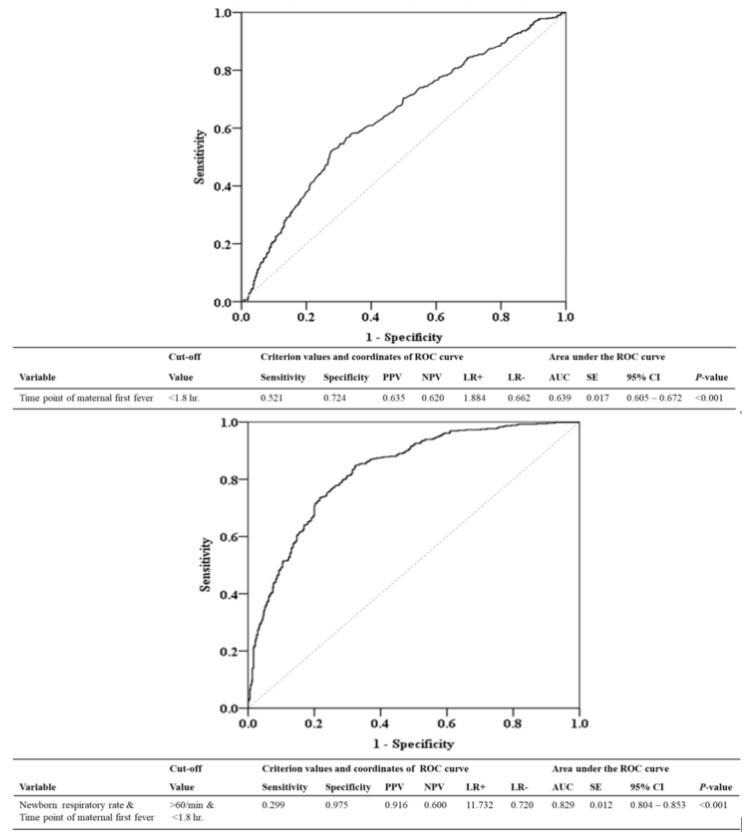
ROC curve to predict whether the baby has probable/proven sepsis based on the time point of the mother’s first fever (baby birth time as the origin) and infant’s respiratory rate. PPV, positive predictive value; NPV, negative predictive value; LR+, likelihood ratio for a positive test; LR−, likelihood ratio for a negative test; AUC, area under curve; SE, standard error; CI, confidence interval; hr, hour; min, minute.

**Table 1 biomedicines-10-02222-t001:** Relationship between perinatal factors and neonatal sepsis.

	Unlikely Sepsis (*n* = 550)			Probable/Proven (*n* = 509)			*p*-Value
	Median	Q1	Q3	Median	Q1	Q3	
**Maternal Factors** Age Parity Prenatal misoprostol administration to fever time (h) Postpartum misoprostol administration to fever time (h) Prenatal epidural analgesia to fever time (h) Fever to birth time (h) From birth to fever time (h) Time point of first fever (h) (birth time as origin) **Baby’s Factors** Gestational age (week) Birth body weight (g) 1 min APGAR score Time of data from birth (h) Heart rate (beats/min) Respiratory rate (breaths/min)	32.00 1.00 17.98 2.25 13.33 3.75 2.75 2.58 38.71 2989.00 8.00 3.37 152.00 55.00	29.00 1.00 10.47 1.50 8.28 1.37 2.08 1.52 37.71 2694.00 8.00 2.38 144.00 51.00	36.00 2.00 28.92 2.78 21.33 10.13 3.63 3.43 39.43 3254.00 8.00 4.75 158.00 58.00	32.00 1.00 17.44 1.81 14.42 3.41 2.40 1.77 38.86 3122.00 8.00 2.82 160.00 62.00	28.00 1.00 11.85 1.25 8.29 1.32 1.55 −0.63 38.00 2850.00 8.00 1.97 156.00 58.00	35.00 1.00 26.58 2.67 23.27 8.27 3.13 2.78 39.57 3378.00 8.00 3.97 163.00 66.00	0.175 <0.001 0.926 0.008 0.170 0.514 <0.001 <0.001 0.001 <0.001 0.001 <0.001 <0.001 <0.001

Q1, percentile 25; Q3, percentile 75; SD, standard deviation; h, hour; g, gram; min, minute. *p*-value from Mann–Whitney U Test.

**Table 2 biomedicines-10-02222-t002:** Maternal fever time point and bacteremia.

	Microorganism	Maternal Fever Time Point	Time Interval between Fever and Birth (Min)
Case 1	*Escherichia coli*	postpartum fever	149
Case 2	*Streptococcus agalactiae*	postpartum fever	142
Case 3	*Streptococcus gallolyticus ssp pasteurianus*	intrapartum fever	29
Case 4	*Bacillus cereus* group	postpartum fever	241

Min: minutes.

**Table 3 biomedicines-10-02222-t003:** Relationship between perinatal factors and neonatal sepsis.

		Unlikely		Probable/Proven		*p*-Value
		*n* (550)	%	*n* (509)	%	
Parity	1 2 3 4 5	365 149 29 6 1	66.4 27.1 5.3 1.1 0.2	338 102 17 2 0	76.2 20.0 3.3 0.4 0.0	0.004
Delivery type	NSD C/S VED	114 179 257	20.7 32.5 46.7	84 116 309	16.5 22.8 60.7	<0.001
GBS screen	Negative Positive Unknown	410 120 20	74.5 21.8 3.6	397 102 10	78.0 20.0 2.0	0.181
Chorioamnionitis	No Suspected Yes	543 6 0	98.9 1.1 0	496 9 4	97.4 1.8 0.8	0.076
Time point of Misoprostol	No use Before delivery After delivery Use after fever	234 141 163 12	42.5 25.6 29.6 2.2	188 177 127 17	36.9 34.8 25.0 3.3	0.005
Epidural analgesia in labor	No Yes Use after fever	215 332 3	39.1 60.4 0.5	87 418 4	17.1 82.1 0.8	<0.001
Intrapartum fever (38–38.9 °C)	No Yes	477 73	86.7 13.3	367 142	72.1 27.9	<0.001
Fever duration—intrapartum	<4 h 4–8 h 8–12 h >12 h	40 18 9 6	54.8 24.7 12.3 8.2	86 25 17 14	60.6 17.6 12.0 9.9	0.656
Intrapartum fever (≥39 °C)	No Yes	538 12	97.8 2.2	480 29	94.3 5.7	0.003
Postpartum fever (38–38.9 °C)	No Yes	63 487	11.5 88.5	129 380	25.3 74.7	<0.001
Fever duration—postpartum	<4 h 4–8 h 8–12 h >12 h	410 62 8 7	84.2 12.7 1.6 1.4	321 44 7 8	84.5 11.6 1.8 2.1	0.842
Postpartum fever (≥39 °C)	No Yes	465 85	84.5 15.5	451 58	88.6 11.4	0.053
Preterm	No Yes	476 74	86.5 13.5	477 32	93.7 6.3	<0.001
APGAR score −1 min	≥7 <7	547 3	99.5 0.5	496 13	97.4 2.6	0.010
PROM > 18 h	No Yes	533 17	96.9 3.1	466 43	91.6 8.4	<0.001
Fever time point	Intrapartum Postpartum Intra- and postpartum	63 477 10	11.5 86.7 1.8	129 367 13	25.3 72.1 2.6	<0.001

NSD, normal spontaneous delivery; C/S, cesarean section; VED, vacuum extract-delivery; PROM, premature rupture of membrane. *p*-value from chi-square or Fisher’s exact tests, as appropriate.

**Table 4 biomedicines-10-02222-t004:** Logistic regression analysis of probable or proven neonatal sepsis.

	Bivariable Analysis (Crude)			Multivariable Analysis (Adjusted)		
	Odds Ratio	95% CI	*p*-Value	Odds Ratio	95% CI	*p*-Value
**Parity**	0.681	0.553–0.839	<0.001			
**Time point of Misoprostol** No use Before delivery After delivery Use after fever	1.000 1.562 0.970 1.763	1.166–2.094 0.718–1.311 0.822–3.784	0.003 0.842 0.145			
**Epidural analgesia in labor** No Yes Use after fever	1.000 3.111 3.295	2.333–4.149 0.722–15.028	<0.001 0.124	1.000 3.398 2.177	2.355–4.902 0.352–13.468	<0.001 0.403
**Delivery type** NSD C/S VED	1.000 0.879 1.632	0.610–1.268 1.177–2.262	0.492 0.003			
**Intrapartum fever (38****–38.9****°C)** No Yes	1.000 2.528	1.848–3.459	<0.001			
**Maternal fever time point** Intrapartum Postpartum Intra- and post-partum fever	1.000 0.376 0.635	0.270–0.523 0.264–1.527	<0.001 0.310	1.000 0.661 0.969	0.439–0.994 0.341–2.750	0.047 0.953
**APGAR score −1 min** ≥7 <7	1.000 4.779	1.354–16.869	0.015			
**PROM >18 h** No Yes	1.000 2.893	1.628–5.142	<0.001			
**Heart rate**	1.148	1.125–1.172	<0.001			
**Respiratory rate**	1.235	1.200–1.270	<0.001	1.230	1.194–1.267	<0.001

CI, confidence interval; NSD, normal spontaneous delivery; C/S, cesarean section; VED, vacuum extract-delivery; PROM, premature rupture of membrane.

## Data Availability

The data used for this study are available from a publicly accessible repository.

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
