# Peer review of "Relationship between Maternal Fever and Neonatal Sepsis: A Retrospective Study at a Medical Center"

_biomedicines, 2022, doi:10.3390/biomedicines10092222_

Round 1

Reviewer 1 Report

It has been a privilege for me to review this study.

I this the manuscript needs English proofread. The inroduction is very  short. it need to be extended and be relevant to the amount of information given in the disscusion part.

The methodology of the study specially in differentiation between postpartum and intrapartum is not clear to me.

Reviewer 2 Report

The manuscript of Huang and coworkers: “Relationship between maternal fever and neonatal sepsis: a retrospective study at a medical center” (biomedicines-1852245) is a very interesting study in the field and should interest a broad readership of your esteemed journal.

The statistical methods are clearly described and facilitate the interpretations of the authors.

The findings of the authors, that newborns will suffer from sepsis after intrapartum/postpartum </=1.8h fever of the mother are very important. In this regard, the discussion section maybe needs attention because the correlation of intrapartum fever and postpartum fever is a bit misleading (see below and please compare (p7, ll 164-166 with p8, ll 183-185).

Especially the correlation of epidural anesthesia and sepsis is of notice. Together with the finding that the monitoring of respiratory rate can be predictive for neonatal sepsis, the value of this study is supportive for caregivers and neonatologists.

I feel that this manuscript should be considered for publication in the present form.

Reviewer 3 Report

Introduction:

Could authors clarify why the concrete topic of this article is clinically interesting?

Would it be more appropriate to state: “Therefore, this study aimed to determine the relationship between maternal postpartum and intrapartum fever and EONS” If not, justify please.

Material and methods:

Page 2, Data collected from the mothers:

- Can authors clarify how chorioamnionitis was defined?

- Could you clarify if your hospital has a stablished guideline for administration of intrapartum antibiotics?

- Could you clarify if your hospital has a stablished guideline for the mothers GBS positive? Could you clarify in text if GBS positive mothers received intrapartum antibiotics?

- Did mothers with intrapartum fever received intrapartum antibiotics? In that case, why is this information not available?

- Full blood count and inflammatory markers (CRP, procalcitonin…) from mothers was not a variable. Justify why not.

- How do you define or calculate “fever duration” in mums? Clarify in text.

- From the text I understand that a baby, clinically well, with a CRP=1.2 mg/dl, was defined as “probable sepsis” baby. Is this right?

Results:

- Correct minimal grammatical issue in table 2 (intrapartum fever: I uppercase or lowercase?

Discussion:

- First paragraph, line 182-183: It may be interesting for readers to discuss those “numerous studies” with your results.

- I would suggest authors to discuss how administration of intrapartum antibiotics could affect to your results.

- In third paragraph, it is stated that epidural analgesia is associated with neonatal sepsis, which is unlikely probable. It would be interesting for readers to better explain these results.

- I am not sure if the “opened” inclusion of babies to the group of “probable sepsis” induced affected on these results.

Round 2

Reviewer 1 Report

All points were fulfilled

Reviewer 3 Report

Thanks, my doubts have been resolved